# Ki67 and PR in Patients Treated with CDK4/6 Inhibitors: A Real-World Experience

**DOI:** 10.3390/diagnostics10080573

**Published:** 2020-08-08

**Authors:** Michela Palleschi, Roberta Maltoni, Sara Ravaioli, Alessandro Vagheggini, Francesca Mannozzi, Francesca Fanini, Francesca Pirini, Maria Maddalena Tumedei, Eleonora Barzotti, Lorenzo Cecconetto, Samanta Sarti, Silvia Manunta, Paola Possanzini, Anna Fedeli, Annalisa Curcio, Mattia Altini, Ugo De Giorgi, Andrea Rocca, Sara Bravaccini

**Affiliations:** 1Istituto Scientifico Romagnolo per lo Studio e la Cura dei Tumori (IRST) IRCCS, 47014 Meldola, Italy; michela.palleschi@irst.emr.it (M.P.); roberta.maltoni@irst.emr.it (R.M.); alessandro.vagheggin@irst.emr.it (A.V.); francesca.mannozzi@irst.emr.it (F.M.); francesca.fanini@irst.emr.it (F.F.); francesca.pirini@irst.emr.it (F.P.); maria.tumedei@irst.emr.it (M.M.T.); eleonora.barzotti@irst.emr.it (E.B.); lorenzo.cecconetto@irst.emr.it (L.C.); samanta.sarti@irst.emr.it (S.S.); silvia.manunta@irst.emr.it (S.M.); anna.fedeli@irst.emr.it (A.F.); mattia.altini@irst.emr.it (M.A.); ugo.degiorgi@irst.emr.it (U.D.G.); andrea.rocca@irst.emr.it (A.R.); sara.bravaccini@irst.emr.it (S.B.); 2Pathology Unit, Morgagni-Pierantoni Hospital, 47121 Forlì, Italy; paola.possanzini@auslromagna.it; 3Breast Surgery Unit, Morgagni-Pierantoni Hospital, 47121 Forlì, Italy; annalisa.curcio@auslromagna.it

**Keywords:** advanced breast cancer, CDK4/6 inhibitors, Ki67, PR

## Abstract

CDK4/6 inhibitors (CDK4/6i) are recommended in patients with estrogen receptor (ER)-positive, HER2-negative advanced breast cancer (ABC). Up to now, no prognostic biomarkers have been identified in this setting. We retrospectively analyzed the expression of progesterone receptor (PR) and Ki67, assessed by immunohistochemistry, in 71 ABC patients treated with CDK4/6i and analyzed the impact of these markers on progression-free survival (PFS). The majority of patients 63/71 (88.7%) received palbociclib, 4 (5.6%) received ribociclib, and 4 (5.6%) received abemaciclib. A higher median value of Ki67 was observed in cases undergoing second-line treatment (*p* = 0.047), whereas the luminal B subtype was more prevalent (*p* = 0.005). In the univariate analysis of the first-line setting, luminal A subtype showed a trend towards a correlation with a longer PFS (*p* = 0.053). A higher continuous Ki67 value led to a significantly shorter PFS. When the interaction between pathological characteristics and line of treatment was considered, luminal B subtype showed a significantly (*p* = 0.043) worse outcome (Hazard Ratio (HR) 2.84; 1.03–7.82 95% Confidence Interval (CI)). PFS in patients undergoing endocrine therapy plus CDK4/6i was inversely correlated with Ki67 expression but not with PR, suggesting that tumor proliferation has a greater impact on cell cycle inhibitors combined with endocrine therapy than PR expression.

## 1. Introduction

Breast cancer (BC) is the most common malignancy and the leading cause of cancer morbidity among women worldwide. About 70% of all BCs are hormone receptor-positive (HR+) and human epidermal growth factor receptor 2-negative (HER2−) [1]. Although metastatic HR+ HER2− BC has a favorable prognosis with respect to other BC subtypes, the outcome of patients with metastatic disease remains poor, with a median overall survival of 36 months [2]. Hormone therapy is the main treatment for these patients and is routinely used in the adjuvant setting but also recommended for patients with advanced disease. However, primary resistance to hormone therapy occurs in around 20% of cases and virtually all patients eventually develop secondary resistance [3].

In the past decade, randomized controlled trials have led to the introduction of several innovative therapeutic strategies into clinical practice consisting of new targeted therapies combined with hormone treatments for both endocrine-sensitive and endocrine-resistant metastatic BC.

Cell cycle deregulation leading to abnormal proliferation is one of the main characteristics of tumor cells [4]. Targeting proteins involved in the deregulation of cell cycle control is a new approach in BC therapy. Pan-cyclin-dependent kinase (CDK) inhibitors, the first generation of CDK inhibitors, have been studied extensively but have been largely abandoned due to their toxicity. CDK4/6 inhibitors (CDK4/6i) are a novel class of small molecule-targeted drugs that selectively inhibit CDK4 and CDK6, the regulators that advance the cell cycle from G1 to S phase. The combination of CDK4 or CDK6 with cyclin D1 forms a cyclin D1–CDK4/6 complex that phosphorylates retinoblastoma protein (pRb), releasing the transcription factor E2F and leading to the transcription of S-phase specific genes and cell cycle progression [5]. Blocking this cyclin D1–CDK4/6-pRb signaling pathway enables CDK4/6i to arrest the proliferation of tumor cells [6].

Preclinical data and clinical trials have demonstrated the promising antitumor effect of CDK4/6i when combined with hormone therapy [7,8]. CDK4/6 inhibition has recently emerged as an effective and feasible strategy to both prevent and overcome endocrine resistance in metastatic HR-positive BC [9,10,11]. The CDK4/6i used in the clinical practice are palbociclib, approved by the US Food and Drug Administration (FDA) in 2015 for the treatment of postmenopausal HR+/HER2− advanced breast cancer (ABC) in combination with hormone treatment (letrozole or fulvestrant); ribociclib; and abemaciclib.

Different randomized trials have shown that the addition of CDK4/6i to conventional hormone therapy can significantly improve the progression-free survival (PFS) and overall survival (OS) of patients with advanced HR+/HER2− BC [12,13,14,15,16,17,18,19,20]. Based on these results, current guidelines recommend CDK4/6i in combination with endocrine therapy as first-line treatment of HR+, HER2− ABC. There is only some controversy about extremely indolent tumors, where endocrine therapy alone could be enough, and very aggressive tumors, where chemotherapy would be preferable. 

Subgroup analyses of phase III studies on CDK4/6i have not identified any robust predictive or prognostic markers that could help the clinicians address the therapeutic choice [21]. This prompted us to retrospectively analyze the prognostic role of the expression of progesterone receptor (PR) and Ki67 (proliferative index), assessed by immunohistochemistry (IHC), in particular on the progression-free survival (PFS) in a real-world population of patients with ABC receiving CDK4/6i as first- or second-line therapy.

## 2. Materials and Methods

### 2.1. Case Series

Out of 97 patients from our institute treated from May 2017 to July 2019 with CDK4/6i for ABC, 48 underwent first-line treatment (23 letrozole and palbociclib, 4 letrozole and ribociclib, 4 letrozole and abemaciclib, and 17 fulvestrant and palbociclib), 23 second-line (fulvestrant and palbociclib), and 26 third- or subsequent-lines. Only 71 patients receiving first- or second-line treatment were considered in the present analysis. Our retrospective study was performed in accordance with the ethical standards laid down in the 1964 Declaration of Helsinki and was approved by the Medical Scientific Committee of Istituto Scientifico Romagnolo per lo Studio e la Cura dei Tumori (IRST) IRCCS and the Ethical Committees of Area Vasta Romagna, Italy (approval number 2836). All patients, with the exception of those who died or were lost to follow up, gave written informed consent for use of their medical record data for retrospective research. 

We retrospectively collected clinical-pathological and treatment response data from patient medical records. estrogen receptor (ER), PR, and Ki67 were assessed by IHC, and HER2 was assessed by IHC or fluorescence in situ hybridization (FISH) in primary tumor samples. PR and Ki67 were analyzed both as continuous and dichotomized variables, i.e., low (<20%) vs. high (≥20%) according to the St. Gallen guidelines [22].

### 2.2. Biomarker Detection

Tumor samples obtained during surgery were fixed in neutral buffered formalin and embedded in paraffin. Four-micron sections were mounted on positive-charged slides for each patient (Bio Optica, Milan, Italy). Biomarker determinations of ER, PR, and Ki67 were performed by immunohistochemistry (IHC) in accordance with European Quality Assurance guidelines, while HER2/neu was evaluated by FISH using the dual color FISH–PathVysion kit (Abbott Molecular, Abbott Park, Illinois, IL, USA) at the Pathology Unit of Morgagni Pierantoni Hospital in Forlì. Immunostaining was performed with the Ventana Benchmark XT system (Ventana Medical Systems, Tucson, AZ, USA) and the Ultraview DAB Detection Kit (Ventana Medical Systems). Confirm anti-ER (clone SP1, Ventana), confirm anti-PR (clone 1E2, Ventana), and Ki67 (clone Mib-1, Dako, Carpinteria, CA, USA) antibodies were used. Sections were automatically counterstained with hematoxylin II (Ventana Medical Systems) for 16 min. External positive and negative controls were always used for each assay. Biomarker positivity was detected and semiquantitatively quantified as the percentage between immunopositive tumor cells and the total number of tumor cells. Two independent observers evaluated all the samples, and any disagreement was resolved by consensus after joint review using a multihead microscope. Hormone receptors, Ki67, and HER2 biomarkers were classified on the basis of the St. Gallen [22] and American Society of Clinical Oncology (ASCO)-College of American Pathologists (CAP) guidelines [23]. In particular, tumors were considered ER-positive when ≥ 1% of immunoreactive cells were detected. The cutoff for PR-positivity was set at ≥ 1% immunoreactive cells. Ki67 was defined as high when the fraction of positively stained cells was ≥ 20% and as low when < 20%. FISH was used preferentially to determine HER2 status and considered positive if the HER2 gene/chromosome 17 centromere ratio was ≥2 or if the average HER2 gene copy number/cell was ≥ 6. In a minority of cases, HER2 was assessed using the HercepTest (DAKO Corporation), which measures the percentage of immunoreactive neoplastic cells defined according to the intensity and completeness of membrane staining and using the 0–3+ recommended scale. Cases scored as 3+ were considered HER2-positive. In cases of equivocal HER2 immunostaining (2+), FISH was performed. 

Luminal BC cases were defined as A (PR ≥ 20% and low Ki67 (<20%)) or B (PR < 20% and/or high Ki67 (≥20%)).

### 2.3. Statistical Analysis

The Mann–Whitney–Wilcoxon test was used to compare continuous variables between patients who received the inhibitor as first- or second-line treatment. An association between categorical variables and first- and second-line patients was inferred with Fisher′s exact test in the case of two categories or its Monte Carlo equivalent when more than two categories were present. Univariate analyses of the impact of PR, Ki67, line of treatment, and BC subtype on PFS were performed using Cox proportional hazard models, reporting hazard ratios (HR) and 95% confidence intervals (95% CI). Figure 1 and Figure 2 show the Kaplan–Meier estimated survival curves for several groupings; whenever possible, median survival times were calculated starting from these.

All reported *p*-values are two-sided, and *p* < 0.05 was considered statistically significant. All statistical analyses were performed using R statistical language (release 3.6.3; R Foundation for Statistical Computing, Vienna, Austria). 

## 3. Results

Of the 71 patients receiving CDK4/6i, 48 were treated in a first-line setting and 23 were treated in a second-line setting. PR and Ki67 were available in 67 and 66 cases, respectively (Table 1). Sixty-three (88.7%) patients received palbociclib, 4 (5.6%) received ribociclib, and 4 (5.6%) received abemaciclib. Fifty-three (75.7%) had ductal carcinoma, and 11 (15.7%) had lobular carcinoma (Table 1). PR was low in 26 patients and high in 41, whereas Ki67 was low in 37 patients and high in 29 (Table 1). There were 24 cases of luminal A tumors and 42 cases of luminal B. Patient and treatment-line characteristics are reported in Table 1. We did not find any significant differences for the analyzed variables between first- and the second-line treatments, with the exception of Ki67 continuous value and BC subtype. A higher median value of Ki67 was observed in cases treated in a second-line setting (*p* = 0.047), where the luminal B subtype was more prevalent (*p* = 0.005) (Table 1).

In the univariate analysis of the first-line setting, luminal A subtype seemed to induce a slightly longer PFS (*p* = 0.053) (Table 2). When considered as dichotomized variables (high/low), PR, Ki67, and subtype (luminal A or B) did not significantly influence the PFS (Table 2). Conversely, when considered as continuous variables (Table 2), PR did not impact the PFS (Figure 1A), whereas a higher Ki67 value led to a significantly shorter PFS (Figure 1B). No significant associations were found when second-line patients were considered (Table 2). 

When the interaction between pathological characteristics and line of treatment was taken into account, luminal B subtype showed a significantly (*p* = 0.043) worse outcome (HR 2.84; 1.03–7.82 95% CI) (Table 3). Furthermore, Kaplan–Meier estimated survival curves showed that luminal A patients have a slightly better PFS (Figure 2).

## 4. Discussion

To the best of our knowledge, this is the first retrospective real-life analysis that highlights the relationship between Ki67 expression and PFS in ABC patients treated with endocrine therapy plus commercially available CDK4/6i. In a large pooled FDA group analysis [21], Gao et al. investigated all the phase 3 randomized BC trials of CDK4/6i plus endocrine therapy to see whether any clinical-pathological ABC subgroups benefited from the addition of a CDK4/6i to endocrine therapy. Results showed that all patient subgroups obtained a similar benefit in first- and second-line settings and that positive estrogen receptor status was the best predictive biomarker of this benefit [21]. However, PR was not considered as a prognostic factor in this pooled analysis. Moreover, the studied population was heterogeneous and may have differed from the general population.

Di Leo et al. performed an interesting exploratory analysis, examining patient and disease characteristics to identify the patients most likely to benefit from abemaciclib and the best time to initiate treatment [24]. The analysis of clinical factors showed that bone-only disease, liver metastases, tumor grade, PR expression, performance status, treatment-free interval from the end of adjuvant endocrine therapy, and time from diagnosis to recurrence had a prognostic value. Patients with poor prognostic factors showed the greatest benefit from the addition of abemaciclib [24].

A high PR expression was associated with prolonged benefit in patients receiving either palbociclib plus fulvestrant or placebo plus fulvestrant in the Paloma-3 trial, independently of the treatment arm [25]. Furthermore, a recent review and meta-analysis by Ramos-Esquivel et al. [26] showed that the addition of CDK4/6i significantly improved PFS, overall response rate and clinical benefit regardless of age, performance status, and disease setting but not race. In fact, Asian patients exhibited a higher benefit in PFS from the experimental treatment, which comprised CDK4/6i and endocrine therapy [26]. In both pooled and subgroup analyses, Ki67 expression was not investigated as a prognostic factor in relation to response to CDK4/6i. Several retrospective and prospective randomized trials have confirmed a consistent relationship between high Ki67 values and poor outcome in patients with BC. In the neoadjuvant setting, many of the studies reported a significant association between high Ki67 levels and response to chemotherapy, measured by clinical or pathological response [27].

However, the relationship between Ki67 and the benefit from chemotherapy is less clear in the adjuvant setting [28]. A number of studies on neoadjuvant endocrine treatment found that treatment-induced alterations in Ki67 predicted response and patient outcome, even after short-term therapy [28]. Few data are available on the value of Ki67 to predict the benefit from chemotherapy or endocrine therapy in metastatic disease. The main limitation of Ki67 is its poor interlaboratory assay reproducibility for the different antibodies and platforms used. Moreover, different cutoffs and scoring systems have been proposed and used over time [29].

We investigated the role of PR and Ki67 expression in ABC patients treated with CDK4/6i in a real-world mono-institutional experience. Our results showed that PFS seemed negatively influenced by high Ki67 expression but not related to PR, suggesting that the effect of the cell cycle inhibitors could be related to tumor proliferation rate rather than to PR expression. Conversely, in a prior work investigating the role of the same biomarkers in patients receiving first-line endocrine therapy alone, we observed that high PR was independently associated with a long time-to-progression whereas Ki67 was not [30]. Whilst a positive correlation between Ki67 and phosphorylated retinoblastoma protein (Rb) has been found in Rb-proficient BC, Rb-negative cases often show high Ki67 levels [31], and albeit mainly ER-negative, they may also include some luminal B tumors. These highly proliferating, Rb-deficient luminal B BC would probably be resistant to CDK4/6i [7].

Our results are similar to those obtained in phase III randomized clinical trials [13,14,15,16,17,19], in particular, in first- and second-line treatments, the median PFSs were 15 months and 12, respectively.

Despite this study being only hypothesis-generating due to the limited sample size, its retrospective nature, and the lack of a control group, our findings reflect a real-world patient population that differs from the cohorts enrolled in the trials considered in recent subgroup and pooled analyses. Our findings on the prognostic role of Ki67 on PFS in this setting are encouraging and worthy of further examination in a prospective study, which would allow to establish its real prognostic value.

In this era of precision oncology and value-based medicine, future research is warranted to confirm whether these or other candidate biomarkers are prognostic and/or predictive of response to CDK4/6i in first-, second-, and further-line settings. Standardization of Ki67 and PR methods is also needed, given that ER-positivity remains the only established biomarker for BC patients who are potential candidates for CDK4/6i.

## Figures and Tables

**Figure 1 diagnostics-10-00573-f001:**
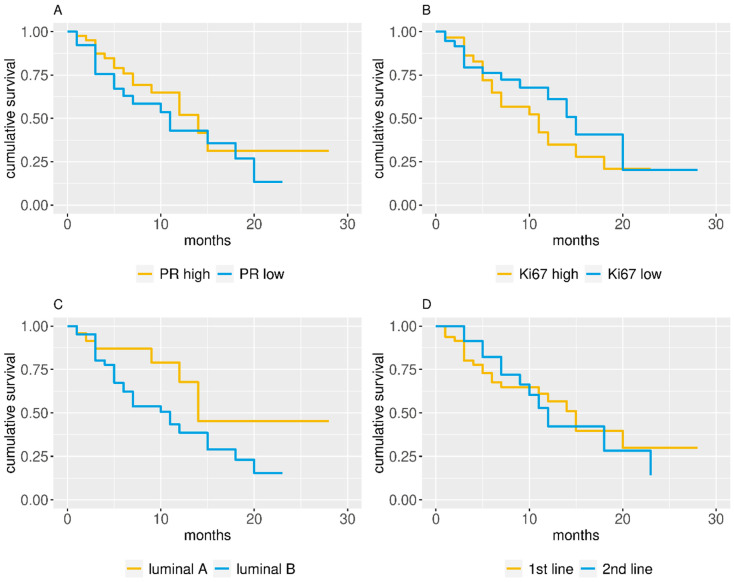
Kaplan–Meier estimated progression-free survival (PFS) curves for (**A**) progesterone receptor (PR) dichotomic value, (**B**) Ki67 dichotomic value, (**C**) cancer subtype, and (**D**) line of treatment.

**Figure 2 diagnostics-10-00573-f002:**
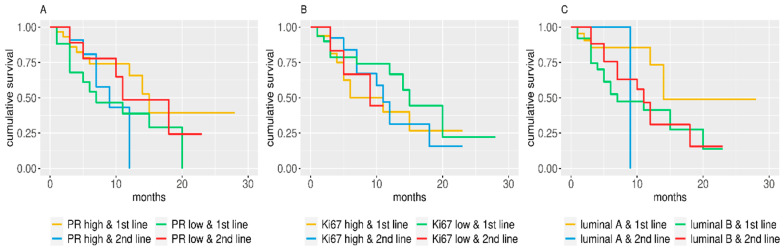
Kaplan–Meier estimated PFS curves for subgroups identified by line of treatment and (**A**) PR dichotomic value, (**B**) Ki67 dichotomic value, and (**C**) and cancer subtype.

**Table 1 diagnostics-10-00573-t001:** Patient characteristics: overall and by line of treatment (results on continuous data are reported in italics).

	Overall		Line I		Line II		
Characteristics	No.	(%)	No.	(%)	No.	(%)	*p*-Value
**Total patients**	71	(100.0)	48	(100.0)	23	(100.0)	
*Patient age at diagnosis*	*55.0 (17.0)*		*56.5 (17.3)*		*53.0 (15.5)*		*0.486* ^a^
*Patient age at start of therapy*	*60.0 (16.5)*		*59.5 (16.3)*		*53.0 (15.5)*		*0.777* ^a^
*Histology*							
Ductal	53	(75.7)	34	(70.8)	19	(83.4)	0.110 ^b^
Lobular	11	(15.7)	9	(18.8)	2	(9.0)	
Other	5	(4.5)	5	(10.4)	0	(0.0)	
Ductal and lobular	1	(1.4)	0	(0.0)	1	(4.5)	
Unknown	1				1		
*Surgery*							
Yes	55	(78.6)	36	(75.0)	19	(86.4)	0.479 ^b^
No	15	(21.4)	12	(25.0)	3	(13.6)	
Unknown	1				1		
*Surgery type* ^c^							
Mastectomy	28	(50.9)	17	(52.8)	11	(57.9)	0.437 ^b^
Conservative	27	(49.1)	19	(47.2)	8	(42.1)	
*PR*	*45.0 (18.5)*		*50.0 (85.5)*		*37.5 (67.5)*		*0.368* ^a^
Low (<20%)	26	(38.8)	17	(36.2)	9	(45.0)	0.431 ^b^
High (≥20%)	41	(61.2)	30	(63.8)	11	(55.0)	
Unknown	4		1		3		
*Ki67*	*15.0 (15.0)*		*15.0 (11.5)*		*25.0 (14.5)*		*0.047* ^a^
Low (<20%)	37	(56.1)	31	(66.0)	6	(31.6)	0.063 ^b^
High (≥20%)	29	(43.9)	16	(34.0)	13	(68.4)	
Unknown	5		1		4		
*Luminal subtype*							
A	24	(36.4)	22	(46.8)	2	(10.5)	0.005 ^b^
B	42	(63.6)	25	(53.2)	17	(89.5)	
Unknown	5		1		4		
*Stage (at diagnosis)*							
I	14	(23.7)	12	(27.9)	2	(12.5)	0.347 ^b^
II	19	(32.2)	11	(25.6)	8	(50.0)	
III	16	(27.1)	12	(27.9)	4	(25.0)	
IV	10	(16.9)	8	(18.6)	2	(12.5)	
Unknown	12		5		7		

^a^ Mann–Whitney–Wilcoxon test for differences in the median with respect to treatment line; ^b^ Fisher exact test for association with line of treatment (histology and stage *p*-values are obtained with 100,000 Monte Carlo replication); ^c^ out of the corresponding patients who underwent surgery.

**Table 2 diagnostics-10-00573-t002:** Univariate analysis.

***First line***	**Characteristics**	**No.**	**Events**	**HR (95%CI)**	**Median PFS (95%CI)**	***p*-Value**
	*PR*					
	Low (<20%)	17	11	2.07 (0.87–4.9)	7.0 (3.0, NA)	0.099
	High (≥20%)	30	10	1.00	15.0 (12.0, NA)	
	*Ki67*					
	Low (<20%)	31	11	1.00	15.0 (12.0, NA)	0.233
	High (≥20%)	16	10	1.69 (0.71–3.99)	8.5 (5.0, NA)	
	*Luminal*					
	A	22	5	1.00	14.0 (14.0, NA)	0.053
	B	25	16	2.72 (0.99–7.48)	7.0 (5.0, NA)	
	*PR (continuous)*			0.99 (0.98–1.01)	-	0.293
	*Ki67 (continuous)*			1.04 (1.01–1.08)	**-**	**0.008**
***Second line***	*PR*					
	Low (<20%)	9	5	1.00	9.0 (7.0, NA)	0.335
	High (≥20%)	11	6	1.88 (0.2, 6.82)	11.0 (10.0, NA)	
	*Ki67*					
	Low (<20%)	6	3	1.50 (0.37, 6.14)	9.0 (5.0, NA)	0.570
	High (≥20%)	13	8	1.00	11.0 (7.0, NA)	
	*Luminal*					
	A	2	1	1.44 (0.17, 12.06)	9.0 (NA, NA)	0.735
	B	17	10	1.00	11.0 (7.0, NA)	
	*PR (continuous)*			1.01 (0.99, 1.02)	-	0.461
	*Ki67 (continuous)*			1.03 (0.97, 1.10)	-	0.346

NA, not applicable.

**Table 3 diagnostics-10-00573-t003:** Survival analysis for pathological characteristics and their interaction with line of treatment.

Characteristics	No.	Events	HR (95%CI)	*p*-Value
*PR*				
Low (<20%)	26	16	1.00	-
High (≥20%)	41	16	0.48 (0.20–1.14)	0.098
Line 1	47	21	1.00	-
Line 2	20	12	0.62 (0.21, 1.78)	0.372
High PR and line 2	11	6	3.24 (0.74, 14.14)	0.119
*Ki67*				
Low (<20%)	37	14	1.00	-
High (≥20%)	29	18	1.76 (0.74, 4.15)	0.200
Line 1	47	21	1.00	-
Line 2	19	12	1.82 (0.50, 6.68)	0.367
High Ki67 and line 2	13	8	0.45 (0.09, 2.22)	0.327
*Luminal*				
A	24	6	1.00	-
B	42	26	2.84 (1.03, 7.82)	0.043
Line 1	47	21	1.00	-
Line 2	19	12	2.71 (0.31, 23.51)	0.366
B and line 2	17	10	0.30 (0.03, 3.03)	0.310

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
