# Peer review of "Ki67 and PR in Patients Treated with CDK4/6 Inhibitors: A Real-World Experience"

_diagnostics, 2020, doi:10.3390/diagnostics10080573_

Round 1

Reviewer 1 Report

Actually, the question of predictive factors for mBC patients who should receive CDK4/6-inhibitor (CDK4/6i) therapy is of high interest in order to reduce the addition of CDK4/6i to those who presumably will benefit.

With this manuscript, the authors provide a rather high number of patients who are treated with endocrine-based therapy including CDK4/6i with clinical data and the histopathological data. Another valuable point is the high standard of the imunhistochemical evaluation as described in the method section.

However, the actual question is prediction of the effect of CDK4/6i. Prognostic Information will not help in clinical decision making.

The question is: Can we predict the benefit from CDK4/6 inhibitors by Ki-67 expression and/or PR positivity.

Answering this question would only be possible by establishing a control group of patients with endocrine therapy only who are matching in distribution of ER/PR/Ki-67 and type of metastases. It would have been possible to look for a group of patients who received chemotherapy or patients who received single Agent endocrine therapy only - may be from historical cohorts.

I would suggest to add such an similar analysis of at least one control Group (single agent endocrine), in order to compare the data and to show a possible predictive value of Ki-67 and PgR.

Reviewer 2 Report

Dear Authors 

The manuscript is very interesting and it is of utmost importance specifically for clinicians. Hormonal positive advanced breast cancers constitute a challenge for oncologists.

One of the limitations of the study, as addressed, is the number of patients enrolled. 

The result section, whether in the manuscript or in the abstract, should be presented in a more clear, organized way.

Few points are highlighted in the attached file, to be corrected.

English language should be revised.

Thank you.

Reviewer 3 Report

This is an important report on the prognostic markers on the effect of CDK4/6 inhibitors. I am actually working on a similar project, from my point of view, the results in this manuscript provide important although not solid evidence for the further selection of patients receiving CDK4/6 inhibitor.
Given the importance and novelty of this work, I recommend it to be published in the Diagnostics journal. However, the authors should refine their presentations in the manuscript, especially for the figures. They should enlarge the font on the figures etc.

Author Response

We thank the reviewer for his interest on our work. We have modified all the figures, according to the reviewer's suggestions. 

Round 2

Reviewer 1 Report

My question and suggestion was:

The question is: Can we predict the benefit from CDK4/6 inhibitors by Ki-67 expression and/or PR positivity.

Answering this question would only be possible by establishing a control group of patients with endocrine therapy only who are matching in distribution of ER/PR/Ki-67 and type of metastases. It would have been possible to look for a group of patients who received chemotherapy or patients who received single Agent endocrine therapy only - may be from historical cohorts.

Reading the revised manuscript I cannot find suggested data of a control group. Without that, predictive impact of any factor cannot be estimated. Thus the manuscript does not provide new information (LumA do better  than LumB, that ist not new).

Author Response

We have underlined in the Introduction and in the Conclusions that prognostic and/or predictive factors of CDK4/6i are still missing in this setting. However, this was a purely exploratory study based on a retrospective cohort of real world clinical practice, which aimed to analyze the role of PR and Ki67.